# The influence of food processing methods on serum parameters, apparent total-tract macronutrient digestibility, fecal microbiota and SCFA content in adult beagles

Xuan Cai[1], Rongrong Liao[1], Guo Chen[2], Yonghong Lu[1], Yiqun Zhao[3]*, Yi Chen[3]*

**1** Institute of Animal Husbandry & Veterinary Science, Shanghai Academy of Agricultural Science, Shanghai, China, **2** Shanghai Agriculture School, Shanghai, China, **3** Shanghai Vocational College of Agriculture and Forestry, Shanghai, China

\* 4525766@qq.com (ZY); 297937577@qq.com (CY)

## Abstract

Food processing methods may influence the health of dogs. However, previous studies have mostly been based on a comparison of several commercial dog foods with different ingredients. In this study, eighteen adult beagles of the same age and health status (assessed by routine blood tests) were used in the experiments. This study analyzed the effects of the following different processing methods: raw, pasteurized, and high temperature sterilization (HTS) made with the same ingredients and nutrients (based on dry matter) on serum parameters, apparent total-tract macronutrient digestibility, fecal microbiota and short-chain fatty acid (SCFA) content in beagle dogs. The data showed, after a test lasting 56-days, the apparent digestibility (ATTD) of protein and fat in HTS food was 91.9%, which was significantly higher ($P < 0.05$) than that in dry food (89.2%, $P < 0.05$). The serum content of triglyceride increased in beagles fed HTS food ($P < 0.05$), and the number of neutrophils in beagles fed raw food and pasteurized food increased significantly ($P < 0.05$), and the platelet count in beagles fed raw food showed an increasing trend compared with the beagles fed HTS food. Different processing methods had an impact on the intestinal microbiota and SCFA of beagles; at least 14 genera were significantly affected by the food produced using different processing methods. In particular, the abundance of *Allprevotella*, *Escherichia-Shigella* and *Turicibacter*, and the total acid content were lower in beagles fed the raw diet, whereas *Streptococcus*, *Collinsella*, *Bacteroides* and *Ruminococcus gnavus* were more abundant following the HTS diet, and *Lactococcus* showed the highest abundance in beagles fed the pasteurized diet. This study showed that dog food produced by different processing methods affected the health of adult beagles.

## 1. Introduction

Over the years, dog food has changed dramatically. It is recognized that dogs are derived from wolves [1]; thus, original dog food would have been raw meat. However, due to close

---

---

**Data Availability Statement:** Raw sequences reads for this study can be found in the NCBI sequence read archive BioProject ID:

PRJNA733866. Other relevant data are within the paper and its Supporting Information files.

**Funding:** This study was supported by fund "Scientific Research Project of Shanghai Vocational College of Agriculture and Forestry" for Yiqun Zhao (grant number KY2-0000-19-01) and "'Run Up' Program for Young Scientists of Shanghai Academy of Agricultural Sciences" for Xuan Cai (grant number ZP19171).

**Competing interests:** The authors have declared that no competing interests exist.

integration with humans over 10,000 years, dogs now accept human carbohydrate-based cooked food [2]. The industrial revolution has changed the form of dog food. Since the invention of puffed dog food, this type of food has rapidly grown. In recent years, some people think that we should return dogs to "natural food", so more and more owners feed their dogs wet food, especially, raw food [3].

There is controversy regarding which form of food is better for pet health. Algya *et al.* [4] analyzed four different processing methods (extruded, high moisture roasted refrigerated, high-moisture grain-free roasted refrigerated, and raw) for producing dog food, and found that "the lightly cooked and raw diets were highly palatable, highly digestible, reduced blood triglycerides, maintained fecal quality and serum parameters, and modified the fecal microbial community of healthy adult dogs." However, Algya *et al.* [4] used different formula food in their study; therefore, it is difficult to judge whether the impact was caused by the processing method itself or due to the change in raw materials or nutrients. The study by Schmidt *et al.* had similar limitations, as different formula food was adopted in the study [5]. Therefore, to eliminate the interference of raw materials and produce diets with the same raw materials and nutrients, it is necessary to conduct an analysis of wet dog food produced using different processing methods.

We hypothesized that the sterilization method at different temperatures had an impact on the characteristics of the food itself, thus affecting the health of dogs. Therefore, this study aimed to analyze the effects of the following different processing methods: raw, pasteurized, and high temperature sterilization (HTS) on serum parameters, apparent total-tract macronutrient digestibility, fecal characteristics and microbiota in adult beagle dogs, in order to provide a basis for improving the processing of dog food and promoting dog health.

## 2. Methods and materials

### 2.1. Diets

The dog food was prepared using rice, chicken breast, corn, sugar beet meal, meat and bone meal, chicken fat and palatability enhancer (Table 1). Rice, corn, sugar beet meal, meat and bone meal were all commercially available finished powders, and the chicken fat and flavoring agents were pasty. The food was prepared in a clean environment. First, chicken breast meat was made into a meat emulsion, mixed with other raw materials, water was added and thoroughly stirred, and the raw food was directly packaged. The sterilized food was sterilized at 80˚C for 20 min and then packaged under aseptic conditions. The HTS food was autoclaved at 121˚C for 30 min and packaged after cooling to room temperature. All foods were divided and

**Table 1. Ingredients in the experimental diets.**

| Ingredients | Amount (%) |
|---|---|
| Broken rice | 14.5 |
| Chicken meat | 32.4 |
| Corn | 3.4 |
| Poultry fat | 3.1 |
| Meat and bone meal | 1.8 |
| Palatability enhancer [1] | 1.5 |
| Beet pulp | 1.0 |
| Water | 42.3 |

[1] Palatability enhancer was mainly made up of chicken liver extract.

**Table 2. Chemical composition of the experimental diets[1].**

| Nutrients | Raw | Pasteurized | HTS[2] | SEM | P-value |
|---|---|---|---|---|---|
| Dry mass (DM) | 34.64 | 34.03 | 34.89 | 8.01 | 0.94 |
| Crude protein (CP) | 30.59 | 30.27 | 30.42 | 0.15 | 0.92 |
| Fat (EE) | 9.44 | 9.35 | 9.51 | 0.22 | 0.83 |
| Ash | 7.54 | 7.63 | 7.40 | 0.35 | 0.74 |
| Crude fiber (CF) | 1.05 | 0.98 | 1.01 | 0.03 | 0.88 |
| Calcium | 1.42 | 1.39 | 1.40 | 0.10 | 0.78 |
| Phosphorous | 1.34 | 1.31 | 1.33 | 0.02 | 0.94 |

All data are measured values. All foods were tested at least 3 times (n = 3).

[1] Data expressed as %DM basis except DM; DM expressed as-fed basis.

[2] HTS: High temperature sterilization.

stored at -20°C. The raw, pasteurized and HTS dog foods used in this experiment were all processed by Shanghai Weita Pet Products Co., Ltd., Shanghai, China.

The dry mass, crude protein, fat (ether extract), ash, crude fiber, total Ca and P contents of the excreta and diet were determined according to the AOAC (1990) 930.15, 984.13, 954.02, 942.05, 962.09, 927.02 and 965.05 methods, respectively (Table 2).

## 2.2. Animals and treatment

The authors confirm that the ethical policies of the journal, as noted on the journal's author guidelines page, have been adhered to and the appropriate ethical review committee approval has been received. The experimental procedures were approved by the Ethics Committee for Research using Laboratory Animals of Shanghai Vocational College of Agriculture and Forestry.

Eighteen healthy beagle dogs (9 male and 9 female) of the same age (3 years) and similar weight were selected from the Training Base of Shanghai Vocational Technical College of Agriculture and Forestry. The test dogs were randomly divided into three groups with 6 dogs in each group according to the principle of half male and female, and fed raw, pasteurized, and HTS food, respectively. The experiment was conducted over 8 weeks, and the dogs were adaptively fed in the kennel for 1 week before the experiment. Each dog was individually raised in a room 2 m × 2 m × 3 m in size. The rooms were separated by cement walls, and each room was equipped with a food bowl, automatic drinking fountain and cotton pads. The temperature in the rooms was maintained at (25.0 ± 3.0°C during the test period. All beagles were fed once a day at 09:00 am, and were fed 750 g each time. Drinking water was always available. The rooms were cleaned at 09:00 am every day.

## 2.3. Blood sample collection and analyses

On day 0 and 56, blood was collected from the beagles for routine analyses of blood and serum metabolite concentrations. About 5 ml of blood from each beagle was collected *via* the forelimb vein in two appropriate vacutainer tubes: one with $K_2EDTA$ added for routine blood analyses, and the other for serum separation. The tubes with $K_2EDTA$ added were analyzed by an automated blood cell counter (BC-2800Vet, Shenzhen Mindray Bio-Medical Electronics Co., Ltd., Shenzhen, China) immediately. The blood samples without anticoagulant were kept at 4°C overnight and then centrifuged at 3,000 × $g$ at room temperature for 10 min to obtain serum, and the serum was then transported to the laboratory for analysis using a clinical chemistry analyzer.

## 2.4. Fecal collection

Fresh fecal samples were collected after 8 weeks of treatment (54–56 d). Total feces excreted during the collection phase were collected from the bottom of the kennel immediately after observing the beagle's bowel behavior. All fecal samples were divided into two, and frozen at -20˚C for further analysis.

## 2.5. Apparent total tract digestibility study

One fecal sample from each beagle was thawed and dried in an air-dry oven at 60˚C for 48 h. The dry mass, crude protein and fat (ether extract) were analyzed as described above. Acid-insoluble ash was used as an indigestible marker, and was analyzed using the method of Cai *et al.* [6].

## 2.6. DNA extraction and 16S rRNA gene processing

The fecal samples used for sequencing analysis were carefully wiped off the surface of the ultra-clean workbench, and about 0.1 g of the sample not exposed to the external environment was selected for DNA extraction. Microbial genomic DNA was extracted from the feces using a QIAamp DNA stool mini kit (Qiagen, Hilden, Germany) according to the manufacturer's instructions. The concentrations and integrity of genomic DNA were verified with a Nano-drop 2000 spectrophotometer and 1.5% agarose gel electrophoresis. The variable region of 16S rRNA V4 was amplified using the universal primer sequence, 343F: 5′-TACGGRAGGCAG CAG-3′ and 798R: 5′-AGGGTATCTAATCCT-3′. Library construction was performed on bar-coded V4 PCR amplicons and sequenced on the Illumina MiSeq PE250 platform (San Diego, CA, USA).

## 2.7. Data processing

Raw sequences were first filtered, and reads with adapter contamination at the ends of the reads, reads <50 bp, and reads with low quality (quality score <20) were removed with the Trimmomatic program [7]. Subsequently, the qualified double-ended raw data were spliced to obtain paired end sequences with a maximum overlap of 200 bp using Flash [8]. The clean tag sequence was then obtained using the split libraries software in QIIME [9] to remove sequences containing N bases in the paired end sequences, single base repeat sequences greater than six, and sequences with a length less than 200 bp. Finally, UCHIME [10] software was used to remove the chimerism in clean tags, and valid tags were obtained for subsequent operational taxonomic unit (OTU) partition. Sequence clustering was subsequently performed with the Vsearch algorithm [11] and clustered into OTUs. The most abundant sequence in each OTU was selected as a representative.

The taxonomy of each OTU was assigned by blasting the representative sequence against the Green genes reference database (Release 13.8, http://greengenes.secondgenome.com/) using the RDP classifier Naive Bayesian classification algorithm [12]. Unknown archaeal or eukaryotic sequences were filtered and removed. Diversity index data were analyzed statistically with analysis of variance, and significant differences between group means were determined with the least significant difference test. These sequence data have been submitted to the GenBank databases under accession number PRJNA733866.

## 2.8. Short-chain fatty acid detection

For SCFA detection, 1 g of fecal sample was diluted with distilled water, homogenized, and centrifuged at $12{,}000 \times g$ for 10 min. Metaphosphoric acid (0.2 ml, 25% w/v) containing

crotonic acid solution was added into 1 ml of supernatant. After storage overnight at -20˚C, the samples were centrifuged for 10 min at 12,000 ×g. The supernatant was filtered through a 0.22 μm filter, and 0.5 μL of filtrate was injected into a gas chromatograph (7890B, Agilent Technologies, CA, USA) equipped with a flame ionization detector and a capillary column (30 m × 0.32 mm × 0.25 μm film thickness). To measure SCFAs, we used crotonic acid as an internal standard (1.077 mg/L). The column, injector, and detector temperatures were 130, 180, and 180˚C, respectively. Hydrogen gas, produced by a gas generator (Parker Chrom Gas, Parker Hannifin Corporation, MN, USA), was used as the carrier gas at a flow rate of 40 mL/min. A standard SCFA mixture containing acetate, propionate, butyrate, isovaleric, and valeric acid was used for calculation.

## 2.9. Statistical analysis

Data on the growth performance (feed intake, body weight), ATTD, routine blood parameters, serum biochemical parameters, and SCFA content were expressed as means ± standard error of the mean (mean ± SEM). All the data mentioned above were validated by a Kolmogorov–Smirnov test and the results showed that all data followed normal distribution. These data were then analyzed by one-way ANOVA followed by post-hoc multiple comparison tests using SPSS v.21.0 statistical software.

The data on 16S RNA processing are described in the "Data processing" section. The statistical differences in the final data were also expressed as mean ± SEM except the relative abundance of the top 10 bacteria at the phylum level (Fig 1A). Fig 1A also describes the relative abundance in each dog. The data on 16S RNA processing was also analyzed by one-way ANOVA followed by post-hoc multiple comparison tests using SPSS v.21.0 statistical software.

A P value < 0.05 was considered significant, whereas 0.05 < P value < 0.10 was considered a tendency.

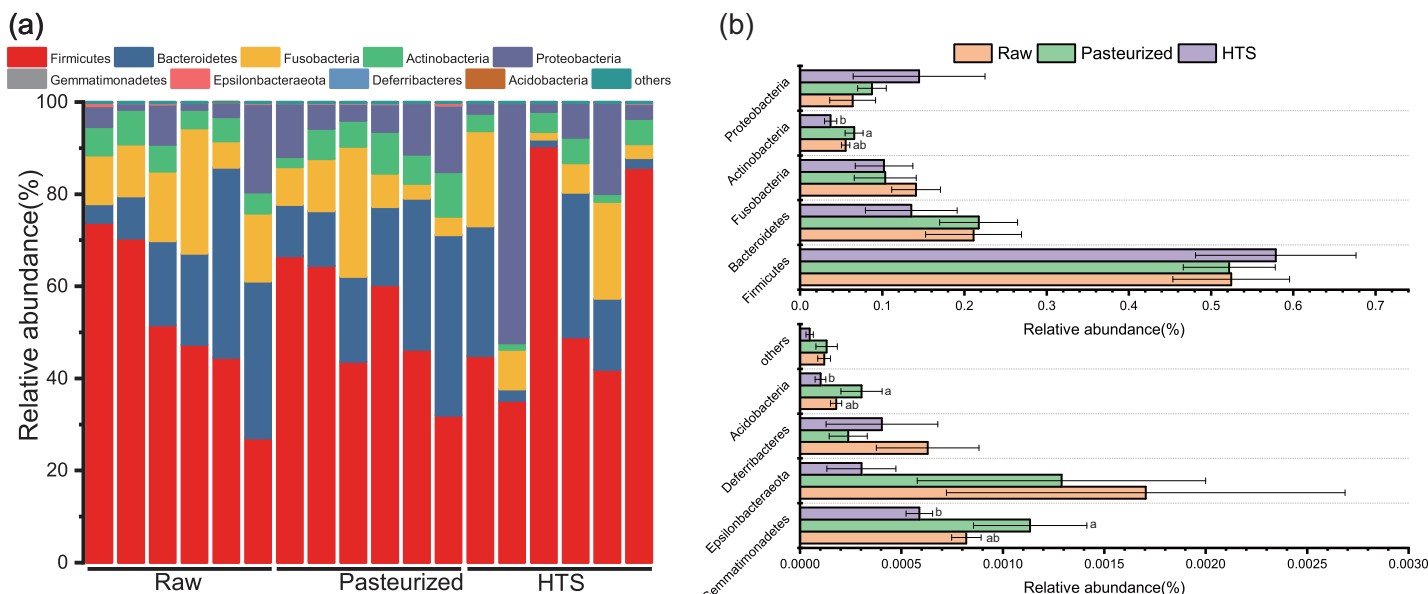

**Fig 1. Distribution of gut microbiome composition and relative change in beagles fed different processed food at the phylum level.** (a) Relative abundance. (b) Relative change in the top 10 bacteria at the phylum level. HTS: High temperature sterilization.

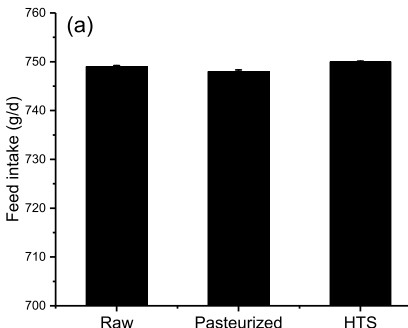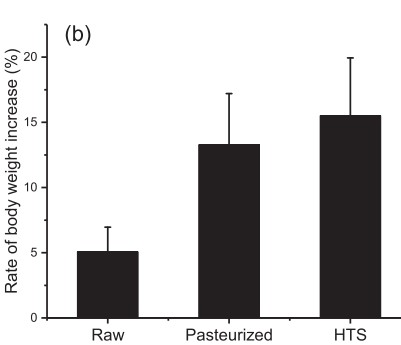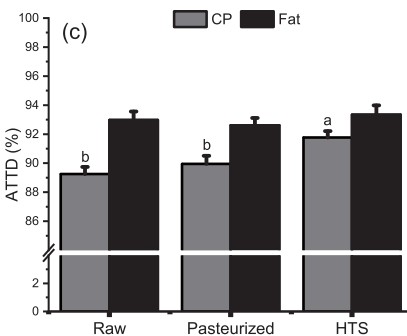

**Fig 2. Effects of food processing methods on food intake, body weight increase (between 0 d and 56 d), and apparent total tract digestibility (ATTD) in adult beagles.** a) Effects of food processing methods on food intake; b) Effects of food processing methods on body weight increase; c) Effects of food processing methods on ATTD. Data are presented as mean ± SEM (n = 6). In each graph, different letters indicated a significant difference using one-way ANOVA ($P < 0.05$). HTS: High temperature sterilization.

## 3. Results

### 3.1. Effects of food processing methods on growth performance and apparent total tract digestibility in adult beagles

Coat color and liveliness in beagle dogs were not changed according to observations by the breeder and other participants in the experiment. As shown in Fig 2A, no differences in food intake were found between the treatments ($P > 0.1$); but all dogs had a significant increase in body weight during the test period ($P<0.05$). Analysis of the growth rate in terms of body weight (Fig 2B), showed that food intake of the different foods did not change, but the dogs fed pasteurized food and HTS food gained weight faster than the dogs fed raw food ($0.05 < P <0.1$). These results showed that the crude protein digestibility of HTS food was significantly higher than that of raw and pasteurized food ($P<0.05$, Fig 2C).

### 3.2. Food processing methods affected serum parameters in adult beagles

No significant differences were found in beagles for both routine blood analysis and serum biochemical parameters at the beginning (0 d, data are shown in S1 & S2 Tables). However, as shown in Table 3, after treatment, the number of neutrophils (Gran) in beagles fed raw food and pasteurized food increased significantly ($P < 0.05$), and the platelet count in beagles fed raw food showed an increasing trend compared with the beagles fed HTS food. The number of white blood cells (WBC), mononuclear cells (Mon) and lymphocytes (Lymph) did not change ($P > 0.1$). The serum biochemical parameters (Table 4) in all dogs after the test were not significantly changed except serum triglyceride (TG). The content of TG increased in beagles fed HTS food compared with dogs fed raw or pasteurized food ($P < 0.05$). Routine blood analysis in all beagles showed normal levels both before and after the test.

### 3.3. Food processing methods affected fecal microbiota composition of adult beagles

In this study, an average of 39, 938 clean tags were obtained for each group, the average length of the sequences was 416 bp, and a mean of 592 observed species was obtained for each group (S3 Table). Rarefaction analysis of 16S rRNA gene sequences indicated adequate sequencing depth (S1 Fig).

Chao1 and observed species indices showed the number of possible taxa, while Shannon and Simpson diversity were used to describe community richness and evenness. These are all

**Table 3. The effect of processing methods on the blood routine of adult beagles.**

| Item | Reference | Raw | Pasteurized | HTS[2] | P-value |
|---|---|---|---|---|---|
| WBC [1] ($10^9 \cdot L^{-1}$) | 6.0–17.0 | 12.42 ± 0.21 | 12.35 ± 0.82 | 11.73 ± 0.35 | 0.608 |
| Lymph ($10^9 \cdot L^{-1}$) | 0.8–5.1 | 3.37 ± 0.33 | 2.85 ± 0.26 | 3.10 ± 0.18 | 0.413 |
| Mon ($10^9 \cdot L^{-1}$) | 0.0–1.8 | 0.92 ± 0.08 | 0.83 ± 0.07 | 0.75 ± 0.03 | 0.207 |
| Gran ($10^9 \cdot L^{-1}$) | 4.0–12.6 | 10.48 ± 0.49a | 10.55 ± 0.68a | 8.38 ± 0.58b | 0.032 |
| RBC ($10^{12} \cdot L^{-1}$) | 5.5–8.5 | 7.31 ± 0.34 | 7.79 ± 0.44 | 7.80 ± 0.27 | 0.549 |
| HGB ($g \cdot L^{-1}$) | 110–190 | 161.83 ± 7.18 | 159.33 ± 11.93 | 160.83 ± 5.79 | 0.979 |
| MCV (fL) | 62–72 | 66.35 ± 1.06 | 67.77 ± 1.16 | 66.88 ± 0.72 | 0.609 |
| PLT ($10^9 \cdot L^{-1}$) | 117–460 | 349.00 ± 17.77A | 342.00 ± 13.23AB | 303.33 ± 9.65B | 0.074 |

Data represent the mean ± standard error of the mean of 6 beagles per treatment (n = 6). Blood from each beagle was collected *via* the forelimb vein in one vacutainer tube containing $K_2$EDTA and instantly analyzed by an automated blood cell counter.

[1]WBC, white blood cell count; Lymph, lymph cell count; Mon, monocyte cell count; Gran, neutrophilic granulocyte count; RBC, red blood cell count; HGB, hemoglobin concentration; MCV, mean corpuscular volume; PLT, platelet count.

[2]HTS: High temperature sterilization.

[a-b] Significant ($P < 0.05$) differences in the same row following analysis with Duncan's test are indicated by different superscript letters. [A-B] A tendency ($P < 0.1$) for differences in the same row following analysis with Duncan's test are indicated by different superscript letters.

alpha diversity indices. Table 5 shows that HTS food decreased both Chao1 and observed species in dogs ($P < 0.05$), and no differences in all alpha diversity indices ($P > 0.1$) were found between raw and pasteurized fed dogs.

*Fusobacterium*, *Streptococcus*, *Lactococcus*, *Prevotella*, and *Alloprevotella* were the top 5 abundant genera in dog feces, which consisted of almost 50% (exactly 48.05%) of the abundance on average in all 502 genera detected in this analysis. The principal component analysis (PCA) plot (S2 Fig) shows microbial communities in the feces of dogs fed food prepared by the different processing methods with no obvious distinction. Table 6 shows at least 14 genera were affected by the food prepared using different processing methods (genera with a relative abundance less than 0.1% were ignored in this analysis). *Streptococcus* was one of the most

**Table 4. The effect of processing methods on serum biochemical parameters in adult beagles.**

| Item | Raw | Pasteurized | HTS[2] | P-value |
|---|---|---|---|---|
| TP[1] ($g \cdot L^{-1}$) | 66.42 ± 1.97 | 64.63 ± 1.54 | 64.25 ± 1.72 | 0.654 |
| GLB ($g \cdot L^{-1}$) | 37.75 ± 1.08 | 35.55 ± 0.76 | 35.27 ± 1.68 | 0.325 |
| ALB ($g \cdot L^{-1}$) | 28.67 ± 1.67 | 29.08 ± 1.32 | 28.98 ± 1.35 | 0.978 |
| TC ($mM \cdot L^{-1}$) | 3.60 ± 0.23 | 4.17 ± 0.45 | 3.75 ± 0.31 | 0.497 |
| TG ($mM \cdot L^{-1}$) | 0.84 ± 0.02b | 0.87±0.03b | 0.99± 0.04a | 0.019 |
| ALT ($U \cdot L^{-1}$) | 39.73 ± 1.20 | 41.68 ± 1.10 | 41.12 ± 1.64 | 0.579 |
| AST ($U \cdot L^{-1}$) | 31.12 ± 1.50 | 31.02 ± 0.81 | 32.62 ± 1.74 | 0.673 |
| Ca ($mM \cdot L^{-1}$) | 2.23 ± 0.14 | 2.44 ± 0.22 | 2.23 ± 0.26 | 0.739 |
| P ($mM \cdot L^{-1}$) | 1.10 ± 0.05 | 1.25 ± 0.06 | 1.16 ± 0.04 | 0.151 |

Data represent the mean ± standard error of the mean of 6 beagles per treatment (n = 6). Blood from each beagle was collected *via* the forelimb vein in one vacutainer tube and the serum analyzed using a clinical chemistry analyzer.

[1]TP, total protein; GLB, globulin; ALB, albumin; TC, total cholesterol; TG, triglyceride; ALT, alanine aminotransferase; AST, aspartate aminotransferase; Ca, calcium content; P, phosphorus content.

[2]HTS: High temperature sterilization.

[a-b] Significant ($P < 0.05$) differences in the same row following analysis with Duncan's test are indicated by different superscript letters.

**Table 5. Alpha diversity indices of the bacterial communities.**

| Item | Raw | Pasteurized | HTS[1] | *P*-value |
|---|---|---|---|---|
| Chao1 | 1024 ± 28ab | 1095 ± 56a | 910 ± 31b | 0.018 |
| observed species | 614 ± 29ab | 657 ± 51a | 515 ± 20b | 0.036 |
| Shannon | 5.0 ± 0.21 | 5.3 ± 0.11 | 4.8 ± 0.28 | 0.272 |
| Simpson | 0.92 ± 0.01 | 0.94 ± 0.01 | 0.91 ± 0.02 | 0.404 |

Data represent the mean ± standard error of the mean of 6 beagles per treatment (n = 6). A fresh fecal sample from each beagle was collected after 8 weeks of treatment. DNA extraction was performed on barcoded V4 PCR amplicons and sequenced on the Illumina MiSeq PE250 platform.

[1] HTS: High temperature sterilization.

[a-b] Significant ($P < 0.05$) differences in the same row are indicated by different superscript letters.

*Firmicutes* (54.17%, means for all samples, and the same below) was the only dominant phylum in the samples from all groups, *Firmicutes*, *Bacteroidetes* (18.79%), *Fusobacteria* (11.57%), *Proteobacteria* (9.88%) and *Actinobacteria* (5.29%) comprised over 99% of the bacterial phyla in the samples from dogs at the beginning of the test (Fig 1A). Fig 1B shows that the abundance of *Actinobacteria*, *Acidobacteria* and *Gemmatimonadetes* in HTS fed dogs decreased compared with those fed pasteurized food ($P < 0.05$).

abundant genera; it increased rapidly in the dogs fed raw food. *Lactococcus* was also one of the main genera in dogs; data showed that the number of *Lactococcus* in dogs fed with pasteurized food increased. *Alloprevotella* was another main genus in dogs; this study showed that the abundance of *Alloprevotella* in dogs fed with HTS food markedly increased. With regard to other bacteria, the abundance of *Collinsella*, *Bacteroides*, *Ruminococcus gnavus*, *Megasphaera*, *Erysipelatoclostridium* and *Lachnospiraceae* decreased in dogs fed HTS food; the abundance of *Escherichia-Shigella* and *Prevotellaceae* decreased in dogs fed raw food, and the abundance of *Ruminococcaceae* increased in dogs fed raw food; the abundance of *Turicibacter* and *Paeniclostridium* increased in dogs fed pasteurized food.

**Table 6. Genus-level taxonomic composition of the bacterial communities.**

| Genera | Raw, % | Pasteurized, % | HTS[1], % | *P*-value[b] |
|---|---|---|---|---|
| Streptococcus | 17.04 ± 4.26a | 6.19 ± 1.95b | 4.11 ± 0.81b | 0.010 |
| Lactococcus | 4.79 ± 1.14b | 15.12 ± 3.37a | 7.93 ± 1.89b | 0.020 |
| Alloprevotella | 6.61 ± 2.06b | 8.11 ± 2.35ab | 16.42 ± 3.7a | 0.054 |
| Collinsella | 4.11 ± 0.53a | 4.44 ± 0.76a | 0.95 ± 0.65b | 0.003 |
| Bacteroides | 3.02 ± 0.85ab | 3.67 ± 1.13a | 1.01 ± 0.33b | 0.097 |
| Escherichia Shigella | 0.45 ± 0.06b | 6.17 ± 1.90a | 4.66 ± 1.82ab | 0.045 |
| Ruminococcus gnavus | 1.47 ± 0.31a | 1.32 ± 0.16a | 0.58 ± 0.09b | 0.018 |
| Turicibacter | 0.53 ± 0.22b | 1.53 ± 0.29a | 0.83 ± 0.11b | 0.015 |
| Megasphaera | 0.34 ± 0.08ab | 0.82 ± 0.37a | 0.04 ± 0.02b | 0.067 |
| Prevotellaceae unclassified | 0.19 ± 0.05b | 0.87 ± 0.27a | 0.68 ± 0.23ab | 0.090 |
| Erysipelatoclostridium | 0.30 ± 0.11a | 0.26 ± 0.03a | ND[2]b | 0.009 |
| Lachnospiraceae | 0.26 ± 0.04a | 0.21 ± 0.04ab | 0.12 ± 0.01b | 0.036 |
| Paeniclostridium | 0.10 ± 0.04b | 0.32 ± 0.10a | 0.14 ± 0.03ab | 0.056 |
| Ruminococcaceae unclassified | 0.19 ± 0.06a | 0.09 ± 0.03ab | 0.05 ± 0.02b | 0.073 |

Data represent the mean ± standard error of the mean of 6 beagles per treatment (n = 6). A fresh fecal sample from each beagle was collected after 8 weeks of treatment. 502 genera were observed, and only the data of relative abundance more than 0.1% and a *P*-value in at least one group less than 0.1 are shown here.

[1] HTS: High temperature sterilization.

[2] ND: The data were less than 0.01, and were defined as "not detected".

[a-b] Significant ($P < 0.05$) differences in the same row are indicated by different superscript letters.

**Table 7. SCFA changes in adult beagles[a].**

| SCFA | RAW | Pasteurized | HTS[2] | P-value |
|---|---|---|---|---|
| Acetic acid (mg·g$^{-1}$) | 3.54 ± 0.37b | 5.22 ± 0.28a | 4.91 ± 0.25a | 0.003 |
| Propionic acid (mg·g$^{-1}$) | 3.59 ± 0.26b | 4.38 ± 0.29a | 4.60 ± 0.17a | 0.027 |
| Butyric acid (mg·g$^{-1}$) | 0.79 ± 0.10b | 1.35 ± 0.20a | 1.53 ± 0.16a | 0.011 |
| Isovaleric acid (mg·g$^{-1}$) | 0.22 ± 0.04 | 0.25 ± 0.03 | 0.31 ± 0.02 | 0.165 |
| Valeric acid (mg·g$^{-1}$) | 0.27 ± 0.09A | 0.28 ± 0.08A | 0.04 ± 0.02B | 0.063 |
| Total acid (mg·g$^{-1}$) | 8.39 ± 0.71b | 11.47 ± 0.69a | 11.38 ± 0.46a | 0.005 |

Data represent the mean ± standard error of the mean of 6 beagles per treatment (n = 6). A fresh fecal sample from each beagle was collected after 8 weeks of treatment. The SCFA in samples were analyzed by gas chromatography.

[1] SCFA, short-chain fatty acids.

[2] HTS: High temperature sterilization.

[a-b] Significant ($P < 0.05$) differences in the same row following analysis with Duncan's test are indicated by different superscript letters. [A-B] A tendency ($P < 0.1$) for differences in the same row following analysis with Duncan's test are indicated by different superscript letters.

## 3.4. Food processing methods changed the gut SCFA composition in adult beagles

Table 7 shows that dog food prepared using different processing methods had a certain effect on the SCFA content in beagle feces. The feces of beagles fed raw food showed significantly lower acetic acid, propionic acid, butyric acid and total acid content ($P < 0.05$); the feces of beagles fed HTS food showed a significantly lower valeric acid content ($P < 0.05$); and the food processing method did not alter isovaleric acid content in beagles ($P > 0.1$).

## 4. Discussion

People's expectations of dog and cat food are very different to those of other animals' food. People no longer care about "production performance indicators" such as daily weight gain and feed-to-meat ratio for dogs and cats, but instead use "health indicators", such as coat color, routine blood analysis, serum parameters, gut health and liveliness. However, the standard for dog and cat health and the food processing method conducive to the health of dogs and cats are controversial.

Weight, coat color, and liveliness are the easiest indicators for dog owners to observe, and they are also the indicators that owners are most concerned about. However, it is difficult to quantitatively describe the coat color and liveliness, and most rely on the subjective feelings of the observer. In this study, the different food processing methods examined had no effect on the coat color and liveliness of beagles. However, the weight of all dogs in the experiment increased significantly ($P < 0.05$), and the weight of beagles fed pasteurized food and HTS food increased more. On the one hand, this change in body weight may be related to higher energy in the food in this study than the food originally consumed by the beagles, and on the other hand, the change may also be related to the difference in digestibility of the foods prepared using different processing methods, although this was not statistically significant ($P > 0.1$). This study demonstrated that the digestibility of protein processing at high temperature was higher.

Routine blood and serum biochemical parameters are commonly used to evaluate animal health. This study also tested routine blood and some routine serum biochemical parameters in beagles. However, it was found that only the number of neutrophils in beagles fed raw and pasteurized food increased, but there was no corresponding significant change in other parameters except platelet count, and it was difficult to determine the cause of this change. The study

by Algya et al. suggested that serum triglyceride content in beagles fed extruded food was significantly higher than that in dogs fed wet food. Our results also showed that HTS food caused a rise in serum triglyceride content in beagles compared with raw or mildly processed (pasteurized) foods.

The intestinal microbiota plays a key role in the efficient absorption and utilization of nutrients, the maintenance of normal intestinal functions, regulation of immune responses, and protection from pathogenic bacteria [13, 14]. Species in the intestinal microbiota may vary greatly. This study showed that *Firmicutes*, *Bacteroidetes*, *Fusobacteria*, *Proteobacteria* and *Actinobacteria* were the main genera in dogs, which was the same as that reported by Paßlack et al. [15]. There have been many studies on the influence of food processing methods on animal intestinal microbiota [4, 16, 17].

As mentioned above, the intestinal microbiota plays a key role in the nutrient metabolism efficiency of the host. At the genus level, the abundance of over 14 genera showed significant differences in beagles fed different diets. Some results in this study were consistent with previous studies. *Alloprevotella* was positively associated with weight, fat mass, and energy metabolism [18]. This may be one of the factors related to the increase in body weight of beagles fed HTS food. *Collinsella* species are usually considered pathobionts as they can affect metabolism by altering intestinal cholesterol absorption, decrease glycogenesis in the liver and increase triglyceride synthesis [19]. This study also showed that the relative abundance of *Collinsella* was positively associated with triglycerides in dogs ($P < 0.05$, by Pearson's analysis). Bermingham et al. [20] reported large reductions in the relative abundance of fecal *Prevotella* in dogs consuming raw-meat diets, and this study also showed that two genera of *Prevotellaceae* decreased in beagles fed raw diets compared to those fed HTS diets. However, some results are not completely consistent: the results of Algya et al. [4] showed that beagles fed raw diets had higher fecal *Lactobacillus*, *Pediococcus* and *Sutterella* compared with those fed pasteurized diets. This study showed no difference in *Lactobacillus* count between beagles fed raw diets and pasteurized diets, while *Lactococcus* count was lower in beagles fed raw diets compared with those fed pasteurized diets. In addition, no significant differences in *Pediococcus* and *Sutterella* in beagles fed different diets were observed. Interestingly, *Streptococcus* is a *Lactobacillales*, and this study showed that it increased in beagles fed pasteurized diets. Due to the huge difference in processing conditions between different diets, it is difficult to judge whether this effect is the result of the processing method itself or other reasons (such as the storage of raw materials, sanitary conditions of the processing environment, etc.)

Some non-starch polysaccharides in food will be fermented by microorganisms to produce SCFAs. In recent years, it has become apparent that SCFAs may play a key role in the prevention and treatment of the metabolic syndrome and bowel disorders [21]. Many intestinal microbiota are related to SCFA in the intestine, such as *Bacteroides*, *Turicibacter*, *Prevotellaceae* etc. Studies have shown that *Turicibacter* was correlated with butyric acid and dietary fiber metabolism [22], and may have an adverse effect on intestinal health [23]. The data from our study showed that the relative abundance of *Turicibacter* in dogs fed pasteurized food significantly increased. Studies have shown that the food processing method could also affect the content of SCFAs [4, 24]. Research by Algya et al. showed that the feces of beagles fed raw food had a higher SCFA content [4], which was confirmed by Sandri et al. [25]. Although this study found that the valeric acid content in the feces of dogs fed raw food increased, the total SCFA content showed the opposite trend. This may be due to the difference in food composition which affected the composition of the intestinal microbiota in beagles, and may in turn have affected the SCFA content in feces. It also shows that there are limits to comparing the effects of two processing methods on animals without guaranteed food ingredients and composition.

Growth performance, apparent total tract digestibility, routine blood analysis, serum parameters, intestinal microbiota structure, and fecal SCFA content often have complex internal links. Thus, it is meaningful to explore more food processing methods for dogs and cats.

This study showed that dog food produced by different processing methods could affect the health of adult beagle dogs. The apparent digestibility of protein and fat in HTS food was higher than that in raw food. The content of TG increased in beagles fed HTS food compared with those fed raw or pasteurized food ($P < 0.05$). The number of neutrophils (Gran) in beagles fed raw food and pasteurized food increased significantly ($P < 0.05$), and the platelet count in beagles fed raw food showed an increasing trend compared with the beagles fed HTS food. Different processing methods had a huge impact on the intestinal microbiota and SCFA in beagles: at least 14 genera were significantly affected by the food produced using different processing methods, but it is difficult to define whether these changes are good or bad. This research provided data on the use of the same raw materials and the same nutritional ingredients to prepare dog food using different processing methods. However, it is limited by the scarcity of experimental animals, sampling conditions, and the lack of mechanism analysis in this study. Experiments with various designs and larger sample sizes are necessary to examine the internal mechanism of dog food processing methods and their effect on dog health.

## Supporting information

**S1 Fig. OTUs detected in this study.** The curve tends to be flat, indicating adequate sampling. (DOCX)

**S2 Fig. Principal component analysis (PCA) plot showing clustering of microbial communities from feces of dogs fed food after different processing methods were not different.** (DOCX)

**S1 Table. Routine blood analysis in adult beagles at the beginning (0 d).** (DOCX)

**S2 Table. The effect of processing methods on serum biochemical parameters in adult beagles (0 d).** (DOCX)

**S3 Table. Quality control of 16S rRNA sequencing.** (DOCX)

**S1 File. Raw data on growth performance and apparent total tract digestibility.** (DOCX)

**S2 File. Raw data on routine blood and serum biochemical parameters.** (DOCX)

**S3 File. Raw data on SCFAs.** (DOCX)

## Acknowledgments

The authors thank Chenwei Zhu, Xingxing Zhang, Yao Ma, Jian Gu, Yuehui Yu, Yiying Tang and Jiayi Shen for their help on beagles feeding and dog food processing.

## Author Contributions

**Conceptualization:** Xuan Cai, Yiqun Zhao, Yi Chen.

**Data curation:** Yiqun Zhao.

**Formal analysis:** Xuan Cai, Rongrong Liao, Yiqun Zhao.

**Methodology:** Xuan Cai, Guo Chen, Yiqun Zhao.

**Project administration:** Xuan Cai, Yiqun Zhao.

**Resources:** Xuan Cai, Yonghong Lu, Yiqun Zhao.

**Validation:** Xuan Cai, Rongrong Liao.

**Visualization:** Xuan Cai, Yiqun Zhao.

**Writing – original draft:** Xuan Cai.

**Writing – review & editing:** Xuan Cai, Rongrong Liao.

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
