## [Editor Report · Decision Letter 0]

21 Jul 2021

PONE-D-21-20446

The influence of food processing methods on serum parameters, apparent total-tract macronutrient digestibility, fecal microbiota and SCFA content in adult beagles

PLOS ONE

Dear Dr. Cai,

Thank you for submitting your manuscript to PLOS ONE. After careful consideration, we feel that it has merit but does not fully meet PLOS ONE’s publication criteria as it currently stands. Therefore, we invite you to submit a revised version of the manuscript that addresses the points raised during the review process.

We look forward to receiving your revised manuscript.

Kind regards,

Alex V Chaves, PhD

Academic Editor

PLOS ONE

Additional Editor Comments:

Manuscript is not formatted for Plos One.

Please follow the instructions provided in our website:

https://journals.plos.org/plosone/s/submission-guidelines

For example, include page numbers and line numbers in the manuscript file. Use continuous line numbers (do not restart the numbering on each page).

Specific details:

Provide specific number of each of the AOAC methods.

Statistical analysis is poor described. Authors must breakdown the statistical analysis for the animal experiment and amplicon sequencing.

For the animal experiment, what were the random and fixed effected used in the model? What was the experimental unit? Were repeated measure analysis used? What is the power for the number of animals used? Why Duncan?

Tables 3 to 5: please provide the P-values Type 3 Tests of Fixed Effects. Defined HTS: Tables must stand alone so any abbreviation must be defined regardless if previously defined.

Table 6: LSMEANS of Erysipelatoclostridium zero? It should be either not detected or increase the decimals.

Manuscript cannot be sent to reviewers as in its current format.

Journal Requirements:

2. To comply with PLOS ONE submissions requirements, in your Methods section, please provide additional information regarding the experiments involving animals and ensure you have included details on (1)housing condition, (2) methods of sacrifice, (3)  amount of  blood taken from each animal  (4) where the blood was taken and (5) methods of anaesthesia if used for blood collection.

---

## [Author Response · Author response to Decision Letter 0]

18 Aug 2021

Revision checklist

1. Manuscript is not formatted for Plos One.

Re: Thank you for your careful check. We have modified the format of this manuscript according to the instructions provided in the website. Although we have tried our best to modify the format of the manuscript, we believe that there are still some format problems in the manuscript. Please point it out in the next revision, and we will correct it seriously.

2. Provide specific number of each of the AOAC methods.

Re: Thank you for your careful check. This has been corrected (line 86-88).

3. Statistical analysis is poor described. Authors must breakdown the statistical analysis for the animal experiment and amplicon sequencing.

Re: Thank you for your careful check. We have rewritten the "statistical analysis" and I don't know if it meets the requirements of the magazine. The data processing in the part of "amplicon sequencing" is described in a section in "Materials and Methods", so the description was not distinguished.

4. For the animal experiment, what were the random and fixed effected used in the model? What was the experimental unit? Were repeated measure analysis used? What is the power for the number of animals used? Why Duncan?

Re: Thank you for your advice. I'm sorry I think we may not fully understand what you mean. I guess your suggestion is for the use of SAS as statistical software for analysis. We are not familiar with SAS software. In this study (except for PCA analysis of sequencing data), all comparisons between groups are based on SPSS software. We did not mention the concept of "random and fixed effected used in the model" in our previous analysis. Correspondingly, we guessed and answered based on the literal meaning. In this study, the health status of the dogs, the raw materials and ingredients of the feed are fixed, and the only variable is the processing method of the feed.

In this study, one dog is a repeat. Therefore, the number of repetitions in this study is six. The supplementary description was added to line 101 and table notes to allow readers to more clearly define the experimental design.

As for why Duncan analysis is used. Well, it seems that there is no absolute standard for which hypothesis testing method to choose. We are based on the results of a Chinese degree thesis (Evalluation of multiple comparison methods with quantitative data): "When the first aim is to explore the difference between the neans of each groups, the LSD and Duncan methods are first selected..." At the same time, the SPPS software expresses the results of Duncan analysis more clearly. Many hypothesis tests in the literature with similar experimental design also use Duncan analysis (Liu et al., 2014. doi: 10.1371/journal.pone.0106412; Song et al., doi: 2017. 10.3382/ps/pex163), so we use Duncan analysis for comparison.

5. Tables 3 to 5: please provide the P-values Type 3 Tests of Fixed Effects. Defined HTS: Tables must stand alone so any abbreviation must be defined regardless if previously defined.

Re: Thank you for your careful check. These have been corrected (Tables 2 to 7, Fig 1 & Fig 2).

6. LSMEANS of Erysipelatoclostridium zero? It should be either not detected or increase the decimals.

Re: Thank you for your careful check. These have been corrected (Line 296).

Re: Thank you for your careful check. These have been corrected.

 

Journal Requirements:

Re: Thank you for your careful check. We have modified the format of this manuscript according to the instructions provided in the website. Although we have tried our best to modify the format of the manuscript, we believe that there are still some format problems in the manuscript. Please point it out in the next revision, and we will correct it seriously.

2. To comply with PLOS ONE submissions requirements, in your Methods section, please provide additional information regarding the experiments involving animals and ensure you have included details on (1)housing condition, (2) methods of sacrifice, (3) amount of blood taken from each animal (4) where the blood was taken and (5) methods of anaesthesia if used for blood collection.

Re: Thank you for your careful check. (1) Housing condition was described in Line 105-108; (2) No animal sacrificed in this study; (3) Amount of blood taken from each animal was described in Line 113; (4) The place blood was taken described in Line 114; (5) Anaesthesia was not used in this study.

Re: Thank you for your advice. Raw sequences reads for this study can be found in the NCBI sequence read archive BioProject ID: PRJNA733866. Other relevant data are within the paper and its Supporting Information files.

---

## [Decision Letter · Decision Letter 1]

5 Oct 2021

PONE-D-21-20446R1The influence of food processing methods on serum parameters, apparent total-tract macronutrient digestibility, fecal microbiota and SCFA content in adult beaglesPLOS ONE

Dear Dr. Cai,

Thank you for submitting your manuscript to PLOS ONE. After careful consideration, we feel that it has merit but does not fully meet PLOS ONE’s publication criteria as it currently stands. Therefore, we invite you to submit a revised version of the manuscript that addresses the points raised during the review process.

We look forward to receiving your revised manuscript.

Kind regards,

Alex V Chaves, PhD

Academic Editor

PLOS ONE

Additional Editor Comments (if provided):

Discussion section has not considered the mechanism for the observed changes. Please fix this on the revised version.

Reviewers' comments:

Reviewer's Responses to Questions

**Comments to the Author**

1. If the authors have adequately addressed your comments raised in a previous round of review and you feel that this manuscript is now acceptable for publication, you may indicate that here to bypass the “Comments to the Author” section, enter your conflict of interest statement in the “Confidential to Editor” section, and submit your "Accept" recommendation.

Reviewer #1: (No Response)

Reviewer #2: (No Response)

2. Is the manuscript technically sound, and do the data support the conclusions?

Reviewer #1: Yes

Reviewer #2: Partly

3. Has the statistical analysis been performed appropriately and rigorously? 

Reviewer #1: Yes

Reviewer #2: Yes

4. Have the authors made all data underlying the findings in their manuscript fully available?

Reviewer #1: Yes

Reviewer #2: Yes

5. Is the manuscript presented in an intelligible fashion and written in standard English?

Reviewer #1: No

Reviewer #2: Yes

6. Review Comments to the Author

Reviewer #1: Additional comments:

PLOS one review: The influence of food processing methods on serum parameters, apparent total tract macronutrient digestibility, fecal microbiota, and SCFA content in adult beagles

The authors in this study examined food processing methods on the health of beagle dogs. Three types of processing methods were assessed to see their resulting effects on the following parameters: apparent total tract digestibility, serum parameters, SCFA content, and fecal microbiota. Overall, a very interesting read with great results. Authors have fixed many of the corrections from the first round of reviews. However, the paper needs to be professionally edited to correct for grammatical, spelling, and punctuation errors. In addition to this, I have included a few minor corrections to improve the quality of the paper.

Major correction:

1. This paper requires a review by a professional editor in order to correct the grammatical, spelling, and punctuation errors. I began to fix some of the mistakes, but they were too numerous to continue. Some of these errors make it difficult to understand the sentences, especially in the materials and methods, and results sections. Therefore, I strongly recommend a professional editor.

Minor corrections:

2. L179: Please provide a brief description validating the use of one-way ANOVA. Ex.: does the data follow parametric assumptions, validated by a KS test (normality) and Levene’s test (homogeneity of variance)?

3. Table 4: The significance letters are not labeled consistently. Specifically, refer to item “TG” under HTS

4. Table 6: In addition to the p-values, please add significance letters to keep it consistent with the other tables.

5. L407-414: Written exactly as in the abstract. Please re-word this.

6. Figure 1c: It would be better to start the y-axis numbering from 0% instead of 80%, and then include a break in the bar graphs instead.

Reviewer #2: The authors have set about testing the question whether there is any nutritional difference between feeding a diet that has been processed in 3 different ways: raw, pasteurised or high temperature sterilization. The premise underlying this project is the statement in the first line of the manuscript: “Food processing methods have a huge influence on the health of dogs.” The experimental design and the methods of analysis are all appropriate for determining any biological effects of feeding to a group of beagle dogs, the same diet processed in 3 different ways. A wide range of assays were undertaken, and statistically different mean values were found in protein digestibility, serum triglycerides, platelet and neutrophil populations in serum and the species range and population proportion of faecal microorganisms of dogs fed on these differently processed diets.

The experimental approach and its technical execution were entirely appropriate to test the hypothesis that the 3 methods of food preparation would show differences in various biological features of the dogs being studied. However, how do the authors justify the findings in terms of the opening sentence of their manuscript: “Food processing methods have a huge influence on the health of dogs.”? What is the evidence that there was any difference between the processing methods and the health of the dogs being studied? Certainly, statistical differences were found in some of the responses being measured, but these differences were so small that it is difficult to see how they would have had any impact on the “health of the dogs”. It is not surprising that heat treatment would affect digestibility of protein. It probably also would have affected starch digestibility in the rice component of the food formulation. Hence the composition of the digestion residue being presented to the microbiome in the large intestine would be different between the 3 processing methods. This would be the likely explanation for the findings of this study. This conclusion should have been pointed out in the Discussion section of the paper, along with an assessment that all of the significantly different changes, observed between the 3 processing methods, would have had negligible effects on the “health” of the dogs. The study was worth doing, but the results are entirely unsurprising.

7. PLOS authors have the option to publish the peer review history of their article (what does this mean?). If published, this will include your full peer review and any attached files.

Reviewer #1: No

Reviewer #2: No

---

## [Author Response · Author response to Decision Letter 1]

25 Oct 2021

Revision checklist

1. This paper requires a review by a professional editor in order to correct the grammatical, spelling, and punctuation errors. I began to fix some of the mistakes, but they were too numerous to continue. Some of these errors make it difficult to understand the sentences, especially in the materials and methods, and results sections. Therefore, I strongly recommend a professional editor.

Re: Thank you for your careful check. We have carefully checked the manuscript and the language of this manuscript has been edited by a company named “International Science Editing” (https://www.internationalscienceediting.com/).

2. L179: Please provide a brief description validating the use of one-way ANOVA. Ex.: does the data follow parametric assumptions, validated by a KS test (normality) and Levene’s test (homogeneity of variance)?

Re: Thank you for your advice. This has been corrected (Line 183-186).

3. Table 4: The significance letters are not labeled consistently. Specifically, refer to item “TG” under HTS

Re: Thank you for your careful check. It has been corrected (Table 4).

4. Table 6: In addition to the p-values, please add significance letters to keep it consistent with the other tables.

Re: Thank you for your careful check. It has been corrected (Table 6, Line 303).

5. L407-414: Written exactly as in the abstract. Please re-word this.

Re: Thank you for your careful check. It has been corrected (Line 412-414)..

6. Figure 1c: It would be better to start the y-axis numbering from 0% instead of 80%, and then include a break in the bar graphs instead.

Re: Thank you for your advice. These have been corrected.

7. The authors have set about testing the question whether there is any nutritional difference between feeding a diet that has been processed in 3 different ways: raw, pasteurised or high temperature sterilization. The premise underlying this project is the statement in the first line of the manuscript: “Food processing methods have a huge influence on the health of dogs.” The experimental design and the methods of analysis are all appropriate for determining any biological effects of feeding to a group of beagle dogs, the same diet processed in 3 different ways. A wide range of assays were undertaken, and statistically different mean values were found in protein digestibility, serum triglycerides, platelet and neutrophil populations in serum and the species range and population proportion of faecal microorganisms of dogs fed on these differently processed diets. 

The experimental approach and its technical execution were entirely appropriate to test the hypothesis that the 3 methods of food preparation would show differences in various biological features of the dogs being studied. However, how do the authors justify the findings in terms of the opening sentence of their manuscript: “Food processing methods have a huge influence on the health of dogs.”? What is the evidence that there was any difference between the processing methods and the health of the dogs being studied? Certainly, statistical differences were found in some of the responses being measured, but these differences were so small that it is difficult to see how they would have had any impact on the “health of the dogs”. It is not surprising that heat treatment would affect digestibility of protein. It probably also would have affected starch digestibility in the rice component of the food formulation. Hence the composition of the digestion residue being presented to the microbiome in the large intestine would be different between the 3 processing methods. This would be the likely explanation for the findings of this study. This conclusion should have been pointed out in the Discussion section of the paper, along with an assessment that all of the significantly different changes, observed between the 3 processing methods, would have had negligible effects on the “health” of the dogs. The study was worth doing, but the results are entirely unsurprising.

Re: Thank you very much for your pertinent evaluation. Health is a difficult concept to define, and different researchers may think differently. Because dogs and cats must be sampled non-destructively, the observable indicators are usually the measurement indicators of serum, feces, and urine, which are usually measured by researchers during the research process, such as PMID: 29893876, 30110340, 33480132. But this article began to define it as health is indeed not rigorous enough, so we have revised it accordingly. 

The prosperity of the pet industry in recent years has brought many novel processing techniques, but the research on the impact of these processing techniques on the health of dogs and cats is still very limited. There have been some studies in the past, but due to processing difficulties, many researchers use different comparing formulas of dry food and wet food, although there are many interesting conclusions, it is difficult to rule out the interference of different formulas and raw materials. This study is trying to use the same formula and the same raw material to process dog food with different characteristics, and analyze its influence on some physiological and biochemical indicators of beagle dogs, and supplement and verify previous studies. In this sense, we think it can help promote the development of the pet food industry. 

However, this study does have some problems. Because of the "schedule" problem of experimental animals, we chose a group test. This resulted in a small number of beagle dogs in each group and large differences within the group, so it is difficult to perform in many analyses significant difference. If we have the opportunity to continue to engage in relevant research, we will be more inclined to use a larger sample size staged trial rather than grouped trials. 

Journal Requirements:

Re: Thank you for your careful check. We have modified the format of this manuscript according to the instructions provided in the website. Although we have tried our best to modify the format of the manuscript, we believe that there are still some format problems in the manuscript. Please point it out in the next revision, and we will correct it seriously.

2. To comply with PLOS ONE submissions requirements, in your Methods section, please provide additional information regarding the experiments involving animals and ensure you have included details on (1)housing condition, (2) methods of sacrifice, (3) amount of blood taken from each animal (4) where the blood was taken and (5) methods of anaesthesia if used for blood collection.

Re: Thank you for your careful check. (1) Housing condition was described in Line 106-109; (2) No animal sacrificed in this study; (3) Amount of blood taken from each animal was described in Line 113; (4) The place blood was taken described in Line 114-115; (5) Anaesthesia was not used in this study.

Re: Thank you for your advice. Raw sequences reads for this study can be found in the NCBI sequence read archive BioProject ID: PRJNA733866. Other relevant data are within the paper and its Supporting Information files.

---

## [Decision Letter · Decision Letter 2]

25 Nov 2021

PONE-D-21-20446R2The influence of food processing methods on serum parameters, apparent total-tract macronutrient digestibility, fecal microbiota and SCFA content in adult beaglesPLOS ONE

Dear Dr. Cai,

Thank you for submitting your manuscript to PLOS ONE. After careful consideration, we feel that it has merit but does not fully meet PLOS ONE’s publication criteria as it currently stands. Therefore, we invite you to submit a revised version of the manuscript that addresses the points raised during the review process.

We look forward to receiving your revised manuscript.

Kind regards,

Alex V Chaves, PhD

Academic Editor

PLOS ONE

Journal Requirements:

Reviewers' comments:

Reviewer's Responses to Questions

**Comments to the Author**

1. If the authors have adequately addressed your comments raised in a previous round of review and you feel that this manuscript is now acceptable for publication, you may indicate that here to bypass the “Comments to the Author” section, enter your conflict of interest statement in the “Confidential to Editor” section, and submit your "Accept" recommendation.

Reviewer #1: All comments have been addressed

Reviewer #2: All comments have been addressed

2. Is the manuscript technically sound, and do the data support the conclusions?

Reviewer #1: Yes

Reviewer #2: Yes

3. Has the statistical analysis been performed appropriately and rigorously? 

Reviewer #1: Yes

Reviewer #2: Yes

4. Have the authors made all data underlying the findings in their manuscript fully available?

Reviewer #1: Yes

Reviewer #2: Yes

5. Is the manuscript presented in an intelligible fashion and written in standard English?

Reviewer #1: Yes

Reviewer #2: Yes

6. Review Comments to the Author

Reviewer #1: (No Response)

Reviewer #2: The authors have satisfactorily changed the wording of their manuscript to avoid implying that the state of health of the dogs was affected by the method of food processing. Certainly some stattstically significant differences were found in some of the measurements being made. These differences were small and would not indicate any change in the state of health of the dogs.

There is one minor correction that is needed. In Tables 6 and 7, mean values plus and minus standard errors are given. The units for these values should be stated, as has been done for the data in the other tables.

7. PLOS authors have the option to publish the peer review history of their article (what does this mean?). If published, this will include your full peer review and any attached files.

Reviewer #1: No

Reviewer #2: No

---

## [Author Response · Author response to Decision Letter 2]

4 Dec 2021

1. The authors have satisfactorily changed the wording of their manuscript to avoid implying that the state of health of the dogs was affected by the method of food processing. Certainly some stattstically significant differences were found in some of the measurements being made. These differences were small and would not indicate any change in the state of health of the dogs.

There is one minor correction that is needed. In Tables 6 and 7, mean values plus and minus standard errors are given. The units for these values should be stated, as has been done for the data in the other tables.

Re: Thank you for your careful check. We corrected this error in Tables 6 and 7 and added units.

---

## [Editor Report · Decision Letter 3]

16 Dec 2021

PONE-D-21-20446R3The influence of food processing methods on serum parameters, apparent total-tract macronutrient digestibility, fecal microbiota and SCFA content in adult beaglesPLOS ONE

Dear Dr. Cai,

Thank you for submitting your manuscript to PLOS ONE. After careful consideration, we feel that it has merit but does not fully meet PLOS ONE’s publication criteria as it currently stands. Therefore, we invite you to submit a revised version of the manuscript that addresses the points raised during the review process. Final remarks from the associate editor:

Did the authors run stats on Table S1? if so please present the P-values.

I just noticed that in Table S2, authors mentioned "Significant (P < 0.05) differences among Duncan analyze in the same row are indicated by different superscript letters." but there are no letters on the LSMEANS. Can you please also add a column with P-values?

S1, S2 & S3 files must be written in English not in Chinese.

We look forward to receiving your revised manuscript.

Kind regards,

Alex V Chaves, PhD

Academic Editor

PLOS ONE
---

## [Author Response · Author response to Decision Letter 3]

19 Dec 2021

1. Did the authors run stats on Table S1? if so please present the P-values.

Re: Thank you for your careful check. We have added a column with P-values. 

2. I just noticed that in Table S2, authors mentioned "Significant (P < 0.05) differences among Duncan analyze in the same row are indicated by different superscript letters." but there are no letters on the LSMEANS. Can you please also add a column with P-values?

Re: Thank you for your careful check. The sentence "Significant (P < 0.05) differences among Duncan analyze in the same row are indicated by different superscript letters." Should deleted here for no significant (P < 0.05) difference existed for the data in this table. We have added a column with P-values in Table S2, and also Table S3. 

3. S1, S2 & S3 files must be written in English not in Chinese.

Re: Thank you for your careful check. We have revised these.

---

## [Editor Report · Decision Letter 4]

21 Dec 2021

The influence of food processing methods on serum parameters, apparent total-tract macronutrient digestibility, fecal microbiota and SCFA content in adult beagles

PONE-D-21-20446R4

Dear Dr. Cai,

We’re pleased to inform you that your manuscript has been judged scientifically suitable for publication and will be formally accepted for publication once it meets all outstanding technical requirements.

Kind regards,

Alex V Chaves, PhD

Academic Editor

PLOS ONE

---

## [Editor Report · Acceptance letter]

27 Dec 2021

PONE-D-21-20446R4 

The influence of food processing methods on serum parameters, apparent total-tract macronutrient digestibility, fecal microbiota and SCFA content in adult beagles 

Dear Dr. Cai:

I'm pleased to inform you that your manuscript has been deemed suitable for publication in PLOS ONE. Congratulations! Your manuscript is now with our production department. 

Kind regards, 

on behalf of

Prof Alex V Chaves 

Academic Editor

PLOS ONE